# Omega-3 Fatty Acids and Exercise in Obesity Management: Independent and Synergistic Benefits in Metabolism and Knowledge Gaps

**DOI:** 10.3390/biology14050463

**Published:** 2025-04-24

**Authors:** Viviana Sandoval, Álvaro Vergara-Nieto, Amanda Bentes, Saulo Silva, Carolina Núñez, Sergio Martínez-Huenchullán

**Affiliations:** 1Carrera de Nutrición y Dietética, Facultad de Ciencias de la Rehabilitación y Calidad de Vida, Universidad San Sebastián, Valdivia 5090000, Chile; saulovnut@gmail.com; 2Departamento de Investigación y Desarrollo, Good Research and Science (GRS), Valdivia 5090000, Chile; contacto@nutrideportiva.cl; 3Escuela de Nutrición y Dietética, Facultad de Ciencias de La Salud, Universidad del Desarrollo, Concepción 4030000, Chile; 4Instituto de Anatomía, Histología y Patología, Facultad de Medicina, Universidad Austral de Chile, Valdivia 5090000, Chile; amandabsimoes@hotmail.com; 5Carrera de Kinesiología, Facultad de Ciencias de la Rehabilitación y Calidad de Vida, Universidad San Sebastián, Valdivia 5090000, Chile; carolina.nunez@uss.cl

**Keywords:** fatty acids, omega-3, exercise, precision medicine

## Abstract

Obesity is a growing global health concern that increases the risk of serious conditions like heart disease, diabetes, and chronic inflammation. In Chile, obesity rates are among the highest worldwide, making it a pressing public health issue. This review explores how omega-3 fatty acids and regular exercise can help manage obesity and improve metabolic health. Omega-3s, commonly found in fish, have been shown to reduce harmful blood fats, lower inflammation, and enhance fat metabolism. Exercise, on the other hand, improves overall fitness, helps regulate blood sugar, and supports cardiovascular health. Different exercise types, such as moderate-intensity continuous training (MICT) and high-intensity interval training (HIIT), provide unique benefits, and their combination with omega-3 supplementation may enhance metabolic adaptations. Additionally, new research suggests that obesity affects the immune system and specific proteins like kallikrein 7 (KLK7), which may interfere with insulin’s ability to regulate blood sugar. While omega-3 and exercise independently show great potential, more research is needed to confirm their combined effects on metabolism, inflammation, and obesity management. Understanding these mechanisms could lead to personalized strategies to combat obesity, improve long-term health outcomes, and optimize exercise performance.

## 1. Obesity and Its Metabolic Consequences

Obesity is a major global public health concern with rising prevalence over recent decades. It contributes to a range of chronic conditions, including cardiovascular, metabolic, respiratory, renal, and cognitive disorders. By 2022, an estimated 16% of adults worldwide lived with obesity, and Chile ranks second among OECD countries with a prevalence of 34.4% [1,2].

The etiology of obesity typically involves an imbalance between energy intake and expenditure, leading to excess white adipose tissue and ectopic fat accumulation in organs like the liver and skeletal muscle. This promotes low-grade systemic inflammation and metabolic dysregulation [3]. Lifestyle interventions—particularly dietary changes and structured physical activity—are recommended for managing obesity. However, the most effective exercise modalities and their mechanisms of action remain under investigation. Obesity also increases the risk of cardiovascular disease (CVD), the leading global cause of death. It contributes to dysregulation of blood pressure, glucose metabolism, and lipid profiles, which in turn promotes atherosclerosis through altered lipid metabolism [4,5].

## 2. Hypertrigliceridemia and the Role of Omega-3 Fatty Acids

Atherosclerosis is a chronic inflammatory disease triggered by the oxidation of low-density lipoproteins (LDL) within the subendothelial space of the arterial wall, leading to low-grade inflammation [6]. High plasma triglyceride levels, a condition known as hypertriglyceridemia, also contribute to the development of cardiovascular diseases (CVDs) through various mechanisms [7,8]. Several pathways highlight the close relationship between hypertriglyceridemia and low-grade inflammatory responses [8]. Elevated triglyceride levels and the lipoproteins that transport them, such as very-low-density lipoproteins (VLDLs) and their remnants, are independently associated with an increased risk of CVD [6,8]. High triglyceride levels are linked to CVD in both fasting and postprandial states, with the postprandial period accounting for most of daily life. Non-fasting triglyceride measurements serve as an independent predictor of various chronic diseases [9], including CVD, such as coronary heart disease [10]. Additionally, other properties of VLDL particles—such as their high triglyceride content, remnant cholesterol, and specific proteins like apolipoprotein C-I and C-III—have been associated with their atherogenic potential and their role in CVD development [11].

Hypertriglyceridemia is a highly prevalent risk factor in the Chilean population, often associated with excess weight. According to the Chilean National Health Survey (2016–2017), the prevalence of overweight and high triglyceride levels (≥150 mg/dL) in adults is 39.8% and 35.8%, respectively, with both rates showing substantial increases since the 2009–2010 survey [12,13]. Lifestyle modifications, particularly a healthy diet, are the primary recommendations for individuals with excess weight and elevated triglycerides. Among these, one of the most effective strategies for reducing both fasting and non-fasting triglyceride levels is increasing the consumption of omega-3 polyunsaturated fatty acids (PUFAs), which have been widely studied for their beneficial effects on human health, particularly in cardiovascular diseases [14]. There are different types of omega-3 fatty acids [15], with three being particularly noteworthy: alpha-linolenic acid (ALA), primarily found in plant sources, and eicosapentaenoic acid (EPA) and docosahexaenoic acid (DHA), both found in marine sources [16,17]. Numerous randomized controlled trials (RCTs) have demonstrated that omega-3, particularly EPA, effectively reduces triglycerides, especially in individuals at high cardiovascular risk. While DHA has been associated with increases in LDL cholesterol, EPA does not exhibit this effect [15]. Recent studies on EPA supplementation (250 mg–2 g/day) have demonstrated significant triglyceride-lowering effects [18,19]. However, the specific impacts of EPA on postprandial triglycerides, low-grade inflammation biomarkers, and VLDL atherogenic properties remain underexplored. Importantly, omega-3 fatty acids influence lipid and inflammatory profiles by regulating the expression of genes involved in lipid metabolism, such as peroxisome proliferator-activated receptors (PPARs) and sterol regulatory element-binding protein 1 (SREBP-1c), which control lipoprotein metabolism and lipid synthesis [20,21].

Insulin resistance and cardiovascular problems are largely driven by chronic low-grade inflammation associated with obesity and metabolic syndrome. Omega-3 fatty acids may help reduce inflammation in adipose tissue and improve insulin sensitivity by promoting a shift in macrophage phenotype from pro-inflammatory (M1) to anti-inflammatory (M2). Additionally, omega-3 polyunsaturated fatty acids (PUFAs) have been shown to reduce systemic inflammation and oxidative stress in metabolic disorders, further supporting their potential as a beneficial dietary intervention for insulin resistance and obesity [22].

## 3. Omega-3 Fatty Acids as Modulator of Inflammation

Inflammation is a complex biological response that plays a dual role in human health—while it is essential for host defense and tissue repair, unresolved chronic inflammation can lead to tissue damage and contribute to various diseases, including cardiovascular diseases (CVDs). The inflammatory process involves intricate interactions between oxidative stress and inflammatory signaling. Reactive oxygen species (ROS) can activate inflammatory pathways, while inflammation itself induces oxidative stress, creating a bidirectional relationship [23]. Omega-3 fatty acids have been shown to reduce plasma levels of inflammatory markers, including tumor necrosis factor-alpha (TNF-α), a cytokine implicated in the pathogenesis of atherosclerosis and other inflammatory diseases. TNF-α is known to induce insulin resistance and dyslipidemia by suppressing genes involved in lipid metabolism and promoting inflammation [24]. Similarly, interleukin-6 (IL-6), a key regulator of inflammatory responses, has been associated with dyslipidemia and metabolic disorders [25]. Elevated triglyceride levels have also been correlated with increased levels of these pro-inflammatory cytokines [26]. Omega-3 fatty acids are recognized as potent anti-inflammatory compounds that may help mitigate diseases characterized by excessive inflammation. Studies suggest that omega-3 supplementation effectively lowers circulating levels of inflammatory cytokines, particularly in patients with heart failure (HF), by reducing TNF-α, IL-6, and C-reactive protein (CRP) levels. This beneficial response to n-3 fatty acid supplementation has been associated with significant reductions in systemic inflammation among HF patients [27].

Inflammation is a complex process involving the production of pro-inflammatory cytokines, chemokines, and lipid mediators such as prostaglandins and leukotrienes, which contribute to immune responses [28,29]. The resolution of inflammation is an active process regulated by specialized pro-resolving mediators (SPMs) derived from omega-3 fatty acids, including resolvins, protectins, and maresins [30,31]. When inflammation fails to resolve properly, it can become chronic, leading to persistent immune activation, sustained production of inflammatory mediators, and tissue damage [32]. Both omega-6 and omega-3 polyunsaturated fatty acids (PUFAs) play crucial roles in this process: omega-6 fatty acids, particularly arachidonic acid, are generally associated with pro-inflammatory effects, whereas omega-3 fatty acids, such as eicosapentaenoic acid (EPA) and docosahexaenoic acid (DHA), have been shown to reduce inflammation by competing with omega-6 pathways, decreasing pro-inflammatory mediator production, and serving as precursors for SPMs [33,34,35]. These findings have fueled growing interest in the therapeutic potential of omega-3 fatty acids, particularly in their ability to regulate inflammatory pathways, support inflammation resolution, and serve as a basis for novel treatments targeting SPMs [35]. Research suggests that omega-3 supplementation may help manage conditions associated with excessive inflammation, such as cardiovascular disease, and may also have potential applications in other inflammatory conditions, including COVID-19 [36]. Maintaining a balance between pro-inflammatory and pro-resolving mediators appears to be a key factor in promoting overall health [31].

A meta-analysis of 38 randomized controlled trials found that EPA monotherapy had a more pronounced effect on reducing cardiovascular mortality than EPA+DHA combined [37]. Omega-3 fatty acids also enhance endothelial function by reducing oxidative stress-induced DNA damage in vascular endothelial cells, thereby contributing to improved vascular health [38].

Several studies indicate that omega-3 polyunsaturated fatty acids (PUFAs) modulate the inflammatory process by altering the nuclear factor-kappa B (NF-κB) and mitogen-activated protein kinase (MAPK) pathways. These signaling pathways regulate the expression of key inflammatory mediators, including tumor necrosis factor-alpha (TNF-α), interleukin-6 (IL-6), and C-reactive protein (CRP). A systematic review and meta-analysis reported that omega-3 supplementation significantly lowered TNF-α and IL-6 levels in patients with heart failure, suggesting a potential role in controlling cardiovascular inflammation. At the molecular level, omega-3 fatty acids modulate inflammation primarily through interference with the nuclear factor kappa B (NF-κB) and mitogen-activated protein kinase (MAPK) signaling cascades [39]. These pathways regulate the transcription of pro-inflammatory genes, including TNF-α, IL-1β, and IL-6, which are central to metabolic inflammation [40]. Moreover, omega-3 fatty acids (12 weeks consuming 2.7 g/day), particularly EPA and DHA, decreased the secretion of IL-1β in human white adipose tissue. It also increased β-cell function and post-prandial fat metabolism [41]. Additionally, omega-3s promote the biosynthesis of specialized pro-resolving mediators (SPMs) such as resolvins (RvE1, RvD1), protectins, and maresins, which not only inhibit further neutrophil infiltration but also enhance macrophage-mediated efferocytosis, leading to the active resolution of inflammation [42]. In addition to the previous findings, epigenetic studies in people with overweight or obesity also show that omega-3s can modulate inflammation and lipid metabolism by altering DNA methylation in genes related to inflammatory and immune response, lipid metabolism, type 2 diabetes, and cardiovascular signaling, such as AKT3, ATF1, HDAC4, and IGFBP5 loci [43]. The anti-inflammatory properties of omega-3 fatty acids are further supported by evidence that EPA and DHA inhibit the activation of toll-like receptors (TLRs), which play a critical role in pathogen recognition and the initiation of inflammatory cascades [44].

## 4. Exercise as a Metabolic Intervention

Physical exercise induces distinct physiological adaptations depending on the prescribed modality, frequency, duration, and intensity, with intensity being a particularly critical factor [45]. Vigorous exercises, characterized by a limited number of repetitions, primarily activate anaerobic pathways to extract energy from glycogen in skeletal muscles. In contrast, lower-intensity exercises, where effort can be sustained for several minutes, rely on nutrient oxidation pathways, utilizing carbohydrates and lipids as energy sources [46]. Given these characteristics, aerobic exercise is frequently recommended for individuals with obesity, particularly due to its beneficial effects on glucose transport and insulin sensitivity under both normal and pathological conditions (e.g., type 2 diabetes). For instance, Kieran et al. demonstrated that just seven days of running or cycling exercise led to significant decreases in fasting insulin levels, improved insulin sensitivity, and enhanced glucose disposal rates in adults with obesity and type 2 diabetes [47].

A common prescription for aerobic exercise is moderate-intensity continuous training (MICT), where individuals maintain a steady effort between 40% and 60% of their heart rate reserve (HRR) throughout the session. Over the past two decades, high-intensity interval training (HIIT) has emerged as an appealing alternative [48]. HIIT involves short bouts of high-intensity effort at 80% to 90% HRR (lasting 1 to 4 min), interspersed with periods of active rest or partial recovery at 10% to 20% HRR. HIIT is often reported to be more enjoyable and time efficient compared to MICT [49]. Some studies suggest that HIIT offers greater physiological adaptations than MICT [48], even in individuals with obesity. However, recent research has challenged this notion. For example, Jung et al. found that after two weeks of MICT and HIIT, adults with prediabetes exhibited similar improvements in cardiorespiratory fitness and systolic blood pressure [48]. Complementing these findings, systematic reviews and meta-analyses have reported comparable outcomes between MICT and HIIT in individuals with obesity, including aerobic capacity, body weight loss, total fat mass, and cholesterol levels [50,51]. Notably, Keating et al. observed that in terms of fat distribution, MICT was more effective than HIIT in reducing android fat percentage in overweight adults [52]. This underscores the possibility that exercise intensity may have specific effects depending on the outcome being measured.

Research on the benefits of exercise for metabolic dysfunctions has shown that combined exercise training (aerobic and anaerobic) [53,54,55,56] or strength-focused exercises [57] can improve physical function. Notably, the aerobic components of these programs are typically of moderate intensity, which consistently leads to improvements in physiological outcomes such as aerobic capacity and reductions in body fat mass. However, studies investigating deeper, tissue-specific metabolic effects of exercise during obesity are limited and predominantly conducted in animal models. These animal studies suggest that the effects of moderate-intensity continuous training (MICT) and high-intensity interval training (HIIT) are tissue dependent. For example, 10 weeks of HIIT led to significant improvements in body insulin sensitivity and increased glucose transporter protein (GLUT4) levels in the gastrocnemius muscle of *db*/*db* mice, changes that were not observed with MICT [58]. Similarly, while both MICT and HIIT demonstrated liver insulin-sensitizing effects in diet-induced obese rats, HIIT uniquely reduced inflammatory mediators such as NF-κB [59]. In white adipose tissue, HIIT also improved insulin signaling in high-fat-fed mice [60]. These findings suggest that HIIT may offer greater metabolic benefits compared to MICT in the context of obesity. However, limitations in exercise prescription methodologies complicate direct comparisons between the two. Most studies normalize exercise intensity by total exercise volume [60] rather than equalizing intensities across programs. This distinction is crucial because when HIIT and MICT are matched for energy expenditure, their metabolic effects on individuals with overweight or obesity, such as reductions in intrahepatic fat levels and circulating insulin, tend to be comparable [61]. It is important to highlight that an alternative type of exercise program has been increasingly implemented in the management of individuals living with obesity. Low-intensity exercise programs have gained popularity among clinicians due to their greater applicability, particularly in people with morbid obesity [62]. Exercise at FATmax intensity—the point at which fat oxidation is maximal—is typically classified as low to moderate intensity and has been shown to effectively enhance fat oxidation [63,64]. However, despite its benefits, this type of training appears to be ineffective in modifying resting fat oxidation and resting energy expenditure, likely due to the low mechanical load involved [64]. Additionally, to date, no studies have specifically investigated the tissue-specific effects of this form of training. To approximate the intensity of physical activity, the talk test scale has been recommended as a practical tool [65]. According to this scale, low-intensity exercise allows individuals to speak comfortably while exercising; moderate-intensity exercise requires pauses for breathing during speech; and high-intensity exercise makes speech difficult, typically limiting it to short phrases. In terms of exercise progression, it is generally recommended to begin with lower intensities to allow adequate adaptation of the cardiorespiratory and musculoskeletal systems, followed by a gradual increase to moderate and/or high intensities as tolerated [66].

In this context, previous experiments conducted by our group demonstrated tissue-specific effects of HIIT and MICT in high-fat-fed mice, where the average intensities, session duration, and distance covered per session were comparable between the two training programs [67].

Based on these observations, we conducted a randomized clinical trial to compare the phenotypical and metabolic effects of 10 sessions of MICT and HIIT in candidates awaiting bariatric surgery. To our knowledge, this was the first study of its kind. Our findings revealed that, while both exercise programs provided similar benefits, such as reductions in waist and hip circumferences and increases in aerobic capacity—most metabolic effects were program-specific. For example, MICT resulted in significant reductions in body weight, body mass index (BMI), and fat mass, along with increases in muscle mass. Furthermore, MICT reduced insulin levels following an oral glucose tolerance test, a change associated with higher protein levels of skeletal muscle PGC-1α, suggesting improved insulin sensitivity through enhanced mitochondrial function in this tissue. In contrast, participants undergoing HIIT exhibited reduced plasma concentrations of FGF21, increased levels of circulatory and white adipose tissue adiponectin (AdipoQ), and decreased liver collagen 1, PGC-1α, and phosphor-AMP-activated kinase (pAMPK)/AMPK ratios. These findings indicate a reduction in liver stress that appears to be uniquely attributed to HIIT [68,69,70] (Figure 1).

These findings led us to hypothesize that exercise intensity exerts specific effects on insulin-sensitive tissues in the context of obesity. However, despite the program-specific benefits observed with MICT and HIIT, no clinical studies have yet investigated the potential mechanisms underlying these differential effects.

## 5. Acute v/s Chronic Effects of Exercise

To gain a deeper understanding of the effects of exercise, particularly in the context of obesity, it is essential to study both the acute responses and chronic adaptations derived from such interventions and how these effects can be modulated by manipulating specific exercise parameters [71]. Previous work from our group demonstrated that varying the intensity of exercise programs results in different acute metabolic effects of MICT and HIIT in physically inactive young adults. For instance, a single session of MICT significantly reduced circulatory levels of FGF21 and insulin [72], while no significant changes in these proteins were observed following a single session of HIIT. These findings raise questions about the potential underlying mechanisms driving these differential effects, which warrant further investigation.

## 6. Exercise and the Immune System

The immune system protects the body against external threats such as viruses, fungi, bacteria, and environmental pollutants. It is divided into two main components: innate and adaptive immunity [73]. The innate immune system serves as the first line of defense and primarily consists of monocytes/macrophages, neutrophils, dendritic cells, and natural killer cells. In contrast, the adaptive immune system, which relies on B and T lymphocytes, can learn and adapt to new threats by recognizing antigens and producing specific antibodies. In recent decades, the role of exercise in modulating immune function has garnered increasing attention. Early findings revealed that athletes were more susceptible to respiratory illnesses (e.g., common colds) following intense competitions. However, individuals with higher physical fitness were observed to have greater protection against upper respiratory tract infections [74]. These early observations sparked interest in understanding the direct effects of exercise on immune cells. For instance, in a wound-healing model using C57BL/6 mice, 10 days of exercise reduced M1 macrophage (pro-inflammatory) infiltration while increasing M2 macrophage (anti-inflammatory) levels. This shift was associated with an accelerated wound-healing rate compared to untrained controls [75]. In humans, evidence remains limited, but recent systematic reviews suggest that both acute and chronic exercise can alter B-cell-related outcomes, such as secretory and plasma immunoglobulin levels. However, the available data are highly heterogeneous regarding populations studied and exercise prescriptions, making it challenging to draw definitive conclusions [76].

In the context of metabolic dysfunctions, the role of the immune system has gained significant attention, as obesity, insulin resistance, and type 2 diabetes are all associated with systemic low-grade inflammation [77]. Notably, exercise has demonstrated a capacity to modulate this inflammatory response. For example, in patients with severe obesity, subcutaneous adipose tissue exhibited macrophage infiltration. However, after a 15-week dietary and exercise intervention, both macrophage infiltration and mRNA levels of inflammatory markers such as IL-6, IL-8, and TNF-α were significantly reduced [78]. Similarly, a mouse model of non-alcoholic steatohepatitis (NASH) showed that 12 weeks of exercise reduced bone marrow-derived macrophage levels in the liver [79]. These findings suggest that the anti-inflammatory effects of exercise extend beyond healthy individuals to those with metabolic dysfunctions, including obesity. However, evidence in human populations remains limited, particularly regarding the comparative effects of different exercise prescriptions on the innate immune system. Exercise modulates immune function through both systemic and tissue-specific effects, many of which are mediated by shifts in immune cell phenotype and cytokine production. One of the most notable mechanisms is the macrophage polarization in adipose tissue, where chronic exercise induces a shift from pro-inflammatory M1 macrophages toward anti-inflammatory M2 phenotypes [80]. Additionally, exercise promotes the mobilization and enhanced function of natural killer (NK) cells [81], changes that could be related to increases in cytotoxic activity and improved immunosurveillance. Chronic moderate exercise also boosts T regulatory (Treg) cell populations, which suppress excessive inflammatory responses and maintain immune tolerance [82]. At the molecular level, exercise increases the expression of peroxisome proliferator-activated receptor gamma coactivator-1 alpha (PGC-1α) in metabolic tissues [83], which supports mitochondrial biogenesis and oxidative metabolism—features that are critical for the anti-inflammatory phenotype of both innate and adaptive immune cells. However, to the best of our knowledge, this area has yet to be thoroughly investigated.

## 7. Cytokines and Exercise

Cytokines, as key mediators of cellular communication, particularly within the immune system, are crucial when studying metabolic dysfunctions. Their dual roles as pro- and anti-inflammatory agents make them valuable biomarkers for assessing the progression or reversal of metabolic impairments. Obesity and insulin resistance are characterized by a low-grade systemic inflammatory state, which is closely linked to immune system dysfunction [84]. A well-known source of this inflammatory state during obesity is the aggregation of immune cells (particularly macrophages) in the white adipose tissue as a result of the expansion of the latter during extended periods of excess energy intake [85]. This leads to a shift in the adipokine profile towards pro-inflammatory markers, including elevated levels of IL-6, TNF, and IL-1β [86]. This imbalance promotes metabolic and functional disruptions in other organs, such as the liver [87] and skeletal muscle. For instance, TNF-α and IL-6 released by hypertrophic adipocytes and infiltrating macrophages can suppress insulin receptor substrate (IRS) phosphorylation, thereby impairing insulin signaling in muscle [88] and liver [89]. Exercise counters this by reducing the expression of these pro-inflammatory cytokines and increasing anti-inflammatory mediators like IL-10 [90]. At a cellular level, exercise alters immune cell metabolism, promoting oxidative phosphorylation over glycolysis in T cells and macrophages [91]—a metabolic switch associated with anti-inflammatory profiles. These adaptations are partly mediated by AMP-activated protein kinase (AMPK) and PGC-1α, both of which are upregulated by exercise and involved in the transcriptional regulation of anti-inflammatory genes and mitochondrial function [92]. Together, these molecular and metabolic shifts contribute to systemic improvements in insulin sensitivity and energy homeostasis. Within this context, the kinin-kallikrein system, traditionally associated with inflammation in coagulation [93], cancer [94], and skin disorders [95], has been implicated in obesity-related metabolic dysregulations [96,97]. Notably, kallikrein 7 (KLK7) has garnered attention due to its role in insulin metabolism and obesity-related phenotypes [98]. For example, KLK7-knockout mice on a high-fat diet (HFD) for 24 weeks exhibited reduced weight gain, lower epididymal fat depots, and decreased levels of leptin and adiponectin—indicators of improved metabolic function—compared to wild-type controls [96]. Additionally, when KLK7 was specifically knocked out in white adipose tissue, transgenic mice fed an HFD showed reduced weight gain, lower circulating leptin levels, and enhanced whole-body insulin sensitivity [97]. Overall, these studies suggest that KLK7 is involved in the metabolic dysfunctions associated with obesity, particularly in terms of insulin action. In that regard, considering the peptidase nature of KLK7, it has been stated that insulin is one of its targets [98], findings that drive us to hypothesize that during obesity, KLK7 increases as part of the adipose tissue dysfunction related to obesity, which in turn decreases the insulin circulatory availability, decreasing the ability of this hormone to induce its expected metabolic functions in their target organs.

Interestingly, the adipokine vaspin has been identified as a KLK7 inhibitor [98]. Elevated circulating vaspin levels in humans have been associated with cardiovascular diseases and related risk factors [99,100]. These findings can be counterintuitive considering the insulin sensitizer function of vaspin [101]; however, it is hypothesized that individuals with obesity may develop compensatory responses leading to resistance to its effects [100]. Evidence suggests that exercise-induced decreases in vaspin levels in individuals with overweight and obesity [102] may enhance vaspin sensitivity and subsequently reduce KLK7 activity, potentially contributing to the metabolic improvements observed with exercise in the context of obesity and insulin resistance. However, clinical studies exploring these mechanisms remain scarce, highlighting the need for further investigation.

## 8. Combination of Omega-3 and Exercise: Potential Benefits and Knowledge Gaps

Omega-3 fatty acids, particularly eicosapentaenoic acid (EPA) and docosahexaenoic acid (DHA), induce key cellular adaptations related to inflammation, energy metabolism, and muscle regeneration. These adaptations include reducing inflammation, improving metabolic efficiency, and promoting muscle repair. Specifically, omega-3 fatty acids modulate cell membranes, enhancing their fluidity and responsiveness to exercise-induced stress. They reduce inflammation by decreasing the activation of nuclear factor kappa B (NF-κB) and lowering the expression of pro-inflammatory cytokines such as tumor necrosis factor-alpha (TNF-α) and interleukin-6 (IL-6). This anti-inflammatory effect is largely mediated through the replacement of pro-inflammatory lipid mediators with specialized pro-resolving mediators such as resolvins and protectins [34]. The combination of omega-3 and exercise has been shown to reduce interleukin-6 (IL-6), TNF-α, and C-reactive protein (CRP) in athletes and individuals with metabolic obesity. Additionally, it has been reported that omega-3 induces a shift in macrophage phenotype from M1 (pro-inflammatory) to M2 (anti-inflammatory), promoting more efficient post-exercise recovery [103]. Recent studies [104] suggest that EPA and DHA supplementation can modify DNA methylation patterns in genes involved in inflammation, cellular aging, and lipid metabolism. Furthermore, findings from the DO-HEALTH trial indicate that omega-3 supplementation, when combined with vitamin D and structured exercise, effectively reduces systemic inflammation and optimizes markers of biological aging, such as PhenoAge and GrimAge [105]. Regarding performance and muscle recovery, Black et al. found that omega-3 supplementation reduced muscle damage and decreased neuromuscular fatigue in rugby players during the preseason [106]. Similarly, Kyriakidou et al. demonstrated that supplementation with 3 g of EPA/DHA improved adaptations to eccentric training (downhill running) by reducing circulating pro-inflammatory cytokines, such as IL-6, and exercise-induced muscle soreness [107]. The anti-inflammatory effects of omega-3 fatty acids have also been explored in other clinical contexts, such as rheumatoid arthritis. A recent systematic review, which analyzed data from over 1000 patients, suggested that omega-3 supplementation may offer some benefits in reducing joint tenderness and improving grip strength. However, the significant heterogeneity among the included clinical trials limits the ability to draw definitive conclusions [108]. Therefore, further high-quality studies are urgently needed to clarify the effects of omega-3 supplementation in rheumatoid arthritis and other musculoskeletal conditions.

At the anabolic level, omega-3 fatty acids modulate the phosphatidylinositol-3-kinase/protein kinase B/mammalian target of rapamycin (PI3K/Akt/mTOR) signaling pathway, promoting muscle protein synthesis by increasing the phosphorylation of p70S6 kinase and 4E-BP1. This effect enhances the anabolic response to resistance training, facilitating muscle regeneration and hypertrophy [109]. In parallel, omega-3 fatty acids activate peroxisome proliferator-activated receptor gamma coactivator 1-alpha (PGC-1α), enhancing mitochondrial biogenesis and the efficiency of oxidative metabolism. This promotes increased utilization of fatty acids as an energy source and reduces muscle tissue dependence on glycogen [110].

Further exploration of the optimal dosage would be valuable, as recent studies indicate beneficial effects within the range of 250 mg to 3 g/day of EPA and DHA, depending on the effects pursued. Two-hundred and fifty milligrams seems enough to decrease circulatory triglyceride levels in seemingly healthy populations [19], and 1 to 3 g for modulating inflammation and enhancing muscle recovery [111]. In this context, it is worth noting a recent systematic review that investigated the potential adverse effects of omega-3 supplementation. After analyzing 90 randomized controlled trials (RCTs), the review reported that individuals who consumed omega-3 supplements had a higher risk of experiencing certain side effects, including diarrhea (25%), dysgeusia (247%), and a tendency to bleed (26%). However, these adverse effects were more commonly observed when the daily intake exceeded 3 g [112]. Despite these findings, the optimal dosage of EPA and DHA to maximize their effects on exercise adaptations remains undefined, since no clear consensus has been reached. Further research is needed to determine their impact on muscle plasticity and their potential to enhance anabolic sensitivity in populations experiencing impaired muscle growth, such as older adults and individuals with sarcopenic obesity. Recent research suggests that supplementation with 1 g of omega-3 fatty acids, comprising 330 mg of eicosapentaenoic acid (EPA) and 660 mg of docosahexaenoic acid (DHA), positively influences aging-related epigenetic pathways, immune responses, and systemic inflammation. Findings from the DO-HEALTH trial, which investigated the effects of daily omega-3 supplementation (1 g/day), vitamin D (2000 IU/day), and a structured home exercise program (SHEP, consisting of 30 min sessions three times per week), revealed that omega-3 supplementation alone slowed biological aging, as assessed by DNA methylation-based clocks, including PhenoAge, GrimAge2, and DunedinPACE. Furthermore, the combination of omega-3 supplementation with vitamin D and structured exercise reduced frailty risk, enhanced longevity, and modulated systemic inflammation and immune function. By reducing oxidative stress, minimizing muscle damage, and improving overall metabolic resilience, omega-3 fatty acids play a crucial role in exercise recovery and physical performance. These anti-inflammatory properties are significant not only for chronic disease prevention but also for improving recovery and enhancing athletic performance [105].

## 9. Clinical Applications and Personalized Intervention Strategies

Considering the literature reviewed here, we summarized the findings and the clinical applications of the studies analyzed:

### 9.1. Omega-3 Supplementation

Recent studies suggest a baseline intake of 250–500 mg/day of EPA+DHA for general cardiovascular health in healthy adults. For individuals with hypertriglyceridemia or metabolic inflammation, doses of 1–3 g/day have shown greater clinical efficacy, particularly for lowering triglycerides and modulating inflammatory markers such as IL-6, TNF-α, and CRP. Intake beyond 3 g/day may increase the risk of side effects, such as gastrointestinal discomfort or bleeding tendencies. Therefore, supplementation should be personalized based on clinical history and monitored periodically.

### 9.2. Exercise Prescription in Obesity Management

For individuals with obesity, the ACSM and WHO recommend at least 150–300 min/week of moderate-intensity aerobic activity (40–60% heart rate reserve (HRR)) or 75–150 min/week of vigorous-intensity exercise (HIIT, 80–90% HRR). Our findings suggest that MICT may be more suitable for patients with visceral obesity and reduced mitochondrial efficiency, while HIIT may offer enhanced benefits for insulin sensitivity and liver function. Low-intensity programs (e.g., FATmax) remain viable options for deconditioned or elderly patients; however, intensity progressions should be considered to achieve the expected metabolic adaptations. Clinical prescription should begin with submaximal testing or a talk test to guide safe and effective intensity thresholds.

### 9.3. Personalization Based on Metabolic Profiles

Because of the previous effects, nutrition and exercise interventions should be personalized depending on the metabolic needs of each person. Aspects such as individual metabolic profiles, including fasting insulin, HOMA-IR, triglyceride levels, and inflammatory cytokines. For instance, individuals with elevated CRP and low adiponectin may benefit from combining omega-3 supplementation with HIIT, while those with sarcopenic obesity may require omega-3-supported resistance training to stimulate anabolic pathways. Integration with multi-omics platforms (e.g., lipidomics, epigenomics) could further refine these recommendations.

## 10. Final Comment

This narrative review emphasizes the potential benefits of personalized omega-3 fatty acid supplementation and tailored exercise interventions as promising synergistic strategies for managing obesity-related metabolic dysfunctions. Obesity remains a critical global health challenge due to its extensive impact on metabolic and cardiovascular health. Both omega-3 fatty acids and structured exercise interventions independently show considerable promise in mitigating these impacts through distinct yet complementary mechanisms. Omega-3 fatty acids, particularly EPA, effectively improve lipid profiles, reduce triglyceride levels, and modulate systemic inflammation by influencing key pathways, including the activation of PPARs and inhibition of pro-inflammatory cytokines. Nevertheless, precise dosage recommendations and long-term outcomes of omega-3 supplementation in diverse populations with obesity remain to be clarified. Exercise interventions such as moderate-intensity continuous training (MICT) and high-intensity interval training (HIIT) also offer substantial metabolic benefits, albeit with differing impacts on specific metabolic markers and tissues. MICT appears particularly effective for improving fat distribution and mitochondrial efficiency, whereas HIIT shows notable advantages for metabolic adaptability and insulin signaling. Such insights underscore the importance of tailored exercise prescriptions based on individual metabolic phenotypes and specific health outcomes. This review further highlights significant knowledge gaps regarding the optimal combination of omega-3 supplementation and exercise. Specifically, future studies should clarify optimal dosing, timing, and the mechanisms underlying the synergistic effects of combined interventions (Figure 2). Additionally, investigating how these strategies influence inflammatory and metabolic outcomes in diverse populations, including individuals with genetic predispositions and different metabolic disorders, is essential. In terms of the limitations of the present report and the literature reviewed, we can highlight the following: (1) Population heterogeneity: We recognize that differences in age, sex, genetic background, metabolic phenotypes, and underlying health conditions (e.g., insulin resistance, cardiovascular disease) may significantly modulate the individual response to omega-3 supplementation and exercise. These inter-individual variations could limit the extrapolation of the findings to broader or more diverse populations; (2) study design limitations: As this is a narrative review, the conclusions drawn rely on the interpretation of existing literature, which includes studies with varying methodologies, intervention durations, and outcome measures. The heterogeneity in clinical trial designs, including inconsistencies in omega-3 dosing, exercise modalities, and endpoints, presents challenges in synthesizing a unified perspective; (3) lack of longitudinal and mechanistic evidence: Despite promising short-term findings, the long-term effects of combined omega-3 and exercise interventions on metabolic health remain insufficiently explored, particularly in human populations. Additionally, the underlying molecular mechanisms—especially the potential synergistic modulation of pathways like NF-κB, PGC-1α, and KLK7—require more robust, mechanistically focused research.

## 11. Conclusions

Integrating omega-3 fatty acid supplementation with personalized exercise interventions could significantly enhance metabolic health, promote recovery, and reduce chronic disease risk. Future research should prioritize well-controlled clinical trials to better define the mechanisms and maximize the therapeutic potential of these combined interventions. This approach will ultimately support the development of precise, evidence-based guidelines for obesity management tailored to individual metabolic profiles, enhancing overall public health outcomes.

## Figures and Tables

**Figure 1 biology-14-00463-f001:**
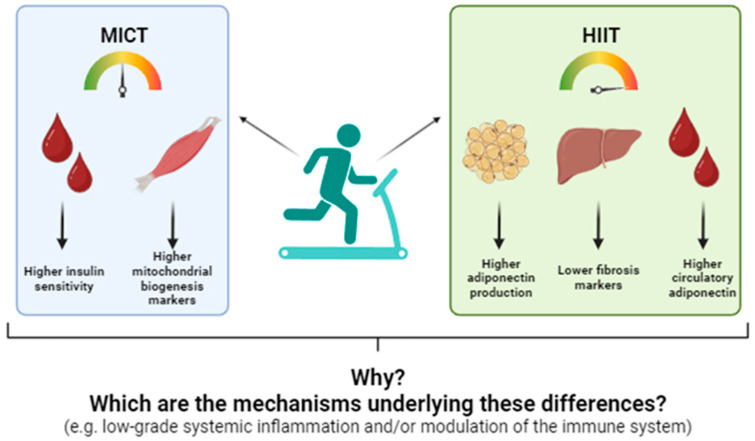
Differential effects of moderate-intensity constant training (MICT) and high-intensity interval training (HIIT) on circulatory and tissue metabolic markers of candidates to undergo bariatric surgery. Adapted from: [70].

**Figure 2 biology-14-00463-f002:**
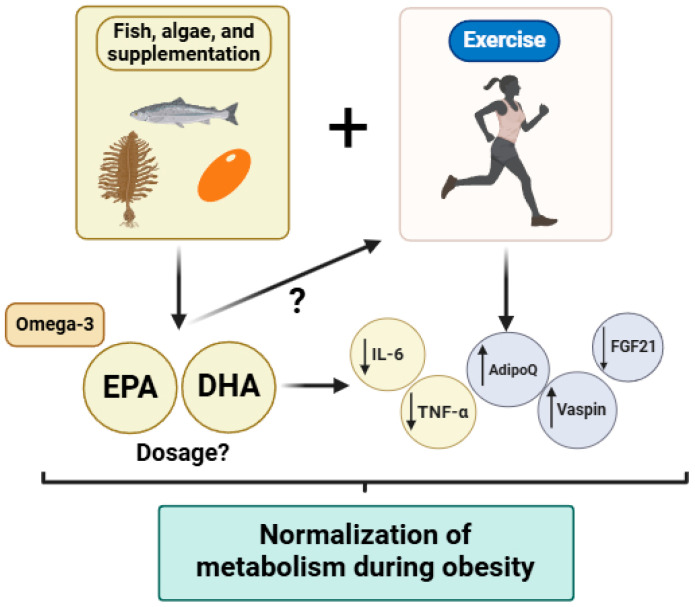
Summary of the synergistic effects of omega-3 fatty acids consumption and the application of exercise programs in the context of metabolic diseases (i.e., obesity, insulin resistance, type 2 diabetes). Abbreviations: EPA: eicosapentaenoic acid; DHA: docosahexaenoic acids; IL: interleukin; TGF-β: transforming growth factor beta; AdipoQ: adiponectin; FGF21: fibroblast growth factor 21. Figure created in BioRender.com “www.biorender.com (accessed on 17 April 2025)”.

## Data Availability

Data are available upon request to the corresponding author at sergio.martinez@uss.cl.

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
