# Peer review of "Omega-3 Fatty Acids and Exercise in Obesity Management: Independent and Synergistic Benefits in Metabolism and Knowledge Gaps"

_biology, 2025, doi:10.3390/biology14050463_

Round 1

Reviewer 1 Report

Comments and Suggestions for Authors

This Sandoval et al, summarizes the interrelated roles of exercise, omega-3 fatty acids, and vitamin D in ameliorating chronic low-grade inflammation associated with obesity. The manuscript is well-structured, describing pathophysiology of obesity and mechanisms of action for various interventions against the low-grade inflammation which can improve metabolic health, promote recovery and decrease chronic disease risks. The authors presented extensive references review of relevant scientific literature that span across basic science, translational research, and multiple randomized controlled trials.

After reviewing “Omega-3 Fatty Acids and Exercise in Obesity Management: 2 Independent and Synergistic Benefits in Metabolism and 3 Knowledge Gaps” from Sandoval et al, I have the following minor comments and suggestions which can further strengthen the manuscript:

  1. Impact on the world population (Section 1: Obesity and its metabolic consequences, p2, p3), lines 61-64, 97-99

The manuscript focuses primarily on Chilean obesity data (e.g., "Chile ranks as the second-highest country within the OECD for obesity prevalence"). Please, if possible, provide global obesity statistics to highlight the importance of obesity worldwide.

  1. Low-Intensity Exercise Training Program (Section 4: Exercise as a metabolic intervention p5, p6 ), lines: 193-214, 242-256

The manuscript discusses MICT and HIIT but could benefit from elaboration of the effect of low-intensity training (i.e. walking). In their discussion of exercise modalities and their tissue-specific effects, authors could kindly address (if any) low-intensity exercise protocols. This information can have benefits for severely obese or elderly individuals who cannot sustain moderate intensity.

  1. Risks of Omega-3 Supplementation (Sections 2 & 8: Hypertrigliceridemia and the role of Omega-3 fatty acids; Omega-3 fatty acids as modulator of inflammation, p3 and p9), lines: 126-146, 388-395

Please include known side effects, contraindications, and upper limits of safe consumption to assess possible risks and contradictions of the omega-3 supplementation (if any).

  1. Injury Recovery Evidence (Section 8: Combination of Omega 3 and exercise, p8, p9), lines: 373-387

This section briefly mentions recovery benefits for muscle damage but doesn't address if there are any evidence of positive effect on the injury recovery (i.e. arthritis or atherosclerosis). If not, please note this as a knowledge gap and future studies should be performed.

  1. Provide Possible Training Plans for low intensity, MICT and HIIT training (Section 4: Exercise as a metabolic intervention, p5), lines: 195-201

While you define MICT (40-60% HRR) and HIIT (80-90% HRR), please clarify how it can be extrapolated on the human population and what type of exercise individuals may perform to correspond to MICT and HIIT. Please include the information on low-intensity training (<40% HRR)

After revieing the manuscript from Sandoval et al, I would recommend it to publication with minor revisions.

Author Response

Dear reviewer,

thank you for your comments and suggestions. Please find attached, point-by-point our reply to each of your concerns.

Kind regards.

Reviewer 2 Report

Comments and Suggestions for Authors

This review provides valuable insights into the independent and synergistic role of omega-3 fatty acids and exercise in obesity management by elaborating on multiple aspects. Overall, the article is informative and well organized.

Here are some of my suggestions, which I hope will help to enhance the academic value of the article.

Major issues:
It is precisely because there are so many points discussed in this paper that the depth of discussion on each of them is relatively limited.
1. The limitations of this study have not been discussed enough.
Although some knowledge gaps are mentioned in the paper, the limitation analysis of the existing studies is not deep enough. For example, in discussing the role of omega-3 fatty acids and exercise in obesity management, the differences that may exist between different populations (e.g., different age, gender, race, disease status, etc.) and the impact of these differences on study results and clinical applications have not been adequately explored. In addition, possible biases in the study (such as selection bias, measurement bias, etc.) are not discussed in detail, which may affect the accuracy and generalizability of the findings.
2. The mechanism is not well explained.
Although the mechanisms of action of omega-3 fatty acids and exercise have been proposed in this paper, some of them are still relatively shallow and lack in-depth analysis at the molecular and cellular biological levels. For example, when discussing the regulatory effect of omega-3 fatty acids on inflammation, although its effect on certain inflammatory signaling pathways is mentioned, the interplay between these pathways and specific molecular regulatory mechanisms is not elaborated. The discussion of the effects of exercise on the immune system also does not delve into how exercise improves metabolic health by modulating a subset of immune cells and their function, which may affect the reader's insight into these mechanisms of action and the development of further research.
3. Lack of detailed discussion of clinical applications
While the article highlighted the potential benefits of omega-3 fatty acids and exercise in obesity management, there was not enough discussion of how to translate these findings into clinical practice. For example, how to determine the optimal dose of omega-3 fatty acids in practical applications, the optimal intensity and frequency of exercise, and how to develop personalized intervention regimens based on an individual's metabolic profile are not detailed. In addition, the application effect of these interventions in different clinical scenarios (such as the application of obesity in patients with cardiovascular disease, diabetes, etc.) has not been thoroughly discussed, which may affect the promotion and application of these research results in clinical practice.
4. Inadequate citations and attention to recently published literature.
Despite the large number of references cited in this paper, some of them were published relatively early and may not fully reflect the latest research progress in this area. In particular, the lack of citations and discussion of recent research findings in some rapidly developing research areas, such as the interplay between exercise and the immune system and epigenetic regulation of omega-3 fatty acids, may make the main points and conclusions of the article seem less cutting-edge and comprehensive.

Minor issues:
1. The information in the introduction is repeated.
Please simplify the introduction, avoid repeating too much background information, make the main points stand out, and improve the flow of reading the article.
2. Inadequate interpretation of technical terms and abbreviations
For professional terms and abbreviations appearing for the first time, full names and simple explanations are given in the text so that the reader can better understand the content of the article.
3. The context of the literature is unclear.
Supplementary tables are recommended to show developmental cues and similarities and differences of similar research literature, so that the reader can better understand the context and trends of related studies.

Author Response

(The authors gave the same response as above.)

Round 2

Reviewer 2 Report

Comments and Suggestions for Authors

Accept.